# SHALLOW DIFFUSE: ROBUST AND INVISIBLE WATERMARKING THROUGH LOW-DIMENSIONAL SUBSPACES IN DIFFUSION MODELS

**Wenda Li**[1*]    **Huijie Zhang**[1*]    **Qing Qu**[1]
[1]Department of Electrical Engineering & Computer Science, University of Michigan
`{wdli,huijiezh,qingqu}@umich.edu`

## ABSTRACT

The widespread use of AI-generated content from diffusion models has raised significant concerns regarding misinformation and copyright infringement. Watermarking is a crucial technique for identifying these AI-generated images and preventing their misuse. In this paper, we introduce *Shallow Diffuse*, a new watermarking technique that embeds robust and invisible watermarks into diffusion model outputs. Unlike existing approaches that integrate watermarking throughout the entire diffusion sampling process, *Shallow Diffuse* decouples these steps by leveraging the presence of a low-dimensional subspace in the image generation process. This method ensures that a substantial portion of the watermark lies in the null space of this subspace, effectively separating it from the image generation process. Our theoretical and empirical analyses show that this decoupling strategy greatly enhances the consistency of data generation and the detectability of the watermark. Extensive experiments further validate that our *Shallow Diffuse* outperforms existing watermarking methods in terms of robustness and consistency.

## 1 INTRODUCTION

Diffusion models (Ho et al., 2020; Song et al., 2021b) have recently become a new dominant family of generative models, powering various commercial applications such as Stable Diffusion (Rombach et al., 2022; Esser et al., 2024), DALL-E (Ramesh et al., 2022; Betker et al., 2023), Imagen (Saharia et al., 2022) Stable Audio (Evans et al., 2024) and Sora (Brooks et al., 2024). These models have significantly advanced the capabilities of text-to-image, text-to-audio, text-to-video, and multi-modal generative tasks. However, the widespread usage of AI-generated content from commercial diffusion models on the Internet has raised several serious concerns: (a) AI-generated misinformation presents serious risks to societal stability by spreading unauthorized or harmful narratives on a large scale (Zellers et al., 2019; Goldstein et al., 2023; Brundage et al., 2018); (b) the memorization of training data by those models (Gu et al., 2023; Somepalli et al., 2023a;b; Wen et al., 2023b; Zhang et al., 2024a) challenges the originality of the generated content and raises potential copyright infringement issues; (c) Iterative training on AI-generated content, known as model collapse (Fu et al., 2024; Alemohammad et al., 2024; Dohmatob et al., 2024; Shumailov et al., 2024; Gibney, 2024) can degrade the quality and diversity of outputs over time, resulting in repetitive, biased, or low-quality generations that may reinforce misinformation and distortions in the wild Internet.

To deal with these challenges, watermarking is a crucial technique for identifying AI-generated content and mitigating its misuse. Typically, it can be applied in two main scenarios: (a) *the server scenario*: where given an initial random seed, the watermark is embedded to the image during the generation process; and (b) *the user scenario*: where given a generated image, the watermark is injected in a post-process manner; (as shown in the left two blocks in Figure 3). Traditional watermarking methods (Cox et al., 2007; Solachidis & Pitas, 2001; Chang et al., 2005; Liu et al., 2019) are mainly designed for the user scenario, embedding detectable watermarks directly into images with minimal modification. However, these methods are vulnerable to attacks. For example, the watermarks can become undetectable with simple corruptions such as blurring on watermarked images. More recent methods considered the server scenario (Zhang et al., 2024c; Fernandez et al., 2023; Wen et al., 2023a; Yang et al., 2024; Ci et al., 2024), where they improve robustness by integrating

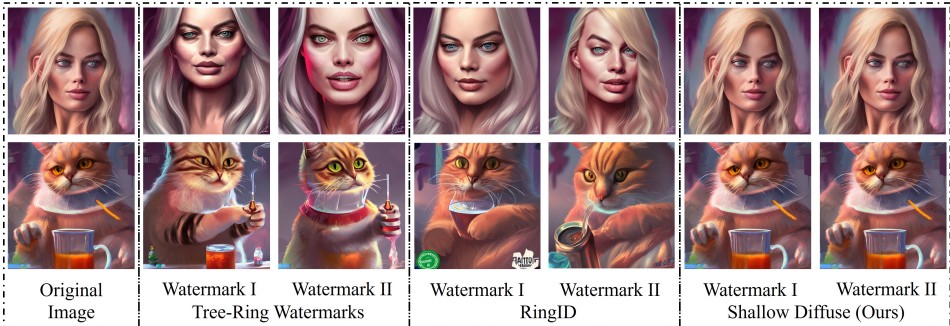

| Original Image | Watermark I | Watermark II | Watermark I | Watermark II | Watermark I | Watermark II |
|---|---|---|---|---|---|---|
| | Tree-Ring Watermarks | | RingID | | Shallow Diffuse (Ours) | |

Figure 1: **Sampling variance of Tree-Ring Watermarks, RingID and Shallow Diffuse.** On the left are the original images, and on the right are the corresponding watermarked images generated using three different techniques: Tree-Ring (Wen et al., 2023a), RingID (Ci et al., 2024), and Shallow Diffuse. For each technique, we generated watermarks using two distinct random seeds, resulting in the respective watermarked images.

watermarking into the sampling process of diffusion models. For example, the work (Ci et al., 2024; Wen et al., 2023a) embeds the watermark into the initial random seed in the Fourier domain and then samples an image from the watermarked seed. As illustrated in Figure 1, these approaches often lead to inconsistent watermarked images because they significantly alter the noise distribution away from Gaussian. Moreover, they require access to the initial random seed, limiting their use in the user scenario. To the best of our knowledge, there is currently no robust and consistent watermarking method suitable for both the server and user scenarios (more detailed discussion about related works could be found in Appendix A).

To address these limitations, we proposed *Shallow Diffuse*, a robust and consistent watermarking approach that can be employed for both the server and user scenarios. Unlike prior works (Ci et al., 2024; Wen et al., 2023a) that embed watermarks into the initial random seed and entangle the watermarking process with sampling, Shallow Diffuse decouples these two steps by leveraging the low-dimensional subspace in the generation process of diffusion models (Wang et al., 2024; Chen et al., 2024). The key insight is that, due to the low dimensionality of the subspace, a significant portion of the watermark will lie in the null space of this subspace, effectively separating the watermarking from the sampling process (see Figure 3 for an illustration). Our theoretical and empirical analyses demonstrate that this decoupling strategy significantly improves the consistency of the watermark. With better consistency as well as independence from the initial random seed, Shallow Diffuse is flexible for both server and user scenarios.

**Our contributions.** The proposed Shallow Diffuse offers several key advantages over existing watermarking techniques (Cox et al., 2007; Solachidis & Pitas, 2001; Chang et al., 2005; Liu et al., 2019; Zhang et al., 2024c; Fernandez et al., 2023; Wen et al., 2023a; Yang et al., 2024; Ci et al., 2024) that we highlight below:

- **Flexibility.** Watermarking via Shallow Diffuse works seamlessly under both server-side and user-side scenarios. In contrast, most of the previous methods only focus on one scenario without a straightforward extension to the other; see Table 1 and Table 2 for demonstrations.

- **Consistency and Robustness.** By decoupling the watermarking from the sampling process, Shallow Diffuse achieves higher robustness and better consistency. Extensive experiments (Table 1 and Table 2 ) support our claims, with extra ablation studies in Figure 4a and Figure 4b .

- **Provable Guarantees.** Unlike previous methods, the consistency and detectability of our approach are theoretically justified. Assuming a proper low-dimensional image data distribution (see Assumption 1), we rigorously establish bounds for consistency (Theorem 1) and detectability (Theorem 2).

## 2 PRELIMINARIES

We start by reviewing the basics of diffusion models (Ho et al., 2020; Song et al., 2021b; Karras et al., 2022), followed by several key empirical properties that will be used in our approach: the low-rankness and local linearity of the diffusion model (Wang et al., 2024; Chen et al., 2024).

### 2.1 PRELIMINARIES ON DIFFUSION MODELS

**Basics of diffusion models.**   In general, diffusion models consist of two processes:

- *The forward diffusion process.* The forward process progressively perturbs the original data $\boldsymbol{x}_0$ to a noisy sample $\boldsymbol{x}_t$ for some integer $t \in [0, T]$ with $T \in \mathbb{Z}$. As in Ho et al. (2020), this can be characterized by a conditional Gaussian distribution $p_t(\boldsymbol{x}_t|\boldsymbol{x}_0) = \mathcal{N}(\boldsymbol{x}_t; \sqrt{\alpha_t}\boldsymbol{x}_0, (1 - \alpha_t)\mathbf{I}_d)$. Particularly, parameters $\{\alpha_t\}_{t=0}^T$ sastify: (*i*) $\alpha_0 = 1$, and thus $p_0 = p_{\text{data}}$, and (*ii*) $\alpha_T = 0$, and thus $p_T = \mathcal{N}(\mathbf{0}, \mathbf{I}_d)$.

- *The reverse sampling process.* To generate a new sample, previous works Ho et al. (2020); Song et al. (2021a); Lu et al. (2022); Karras et al. (2022) have proposed various methods to approximate the reverse process of diffusion models. Typically, these methods involve estimating the noise $\boldsymbol{\epsilon}_t$ and removing the estimated noise from $\boldsymbol{x}_t$ recursively to obtain an estimate of $\boldsymbol{x}_0$. Specifically, One sampling step of Denoising Diffusion Implicit Models (DDIM) Song et al. (2021a) from $\boldsymbol{x}_t$ to $\boldsymbol{x}_{t-1}$ can be described as:

$$\boldsymbol{x}_{t-1} = \sqrt{\alpha_{t-1}} \underbrace{\left( \frac{\boldsymbol{x}_t - \sqrt{1 - \alpha_t}\boldsymbol{\epsilon_\theta}(\boldsymbol{x}_t, t)}{\sqrt{\alpha_t}} \right)}_{:=\boldsymbol{f_{\theta,t}}(\boldsymbol{x}_t)} + \sqrt{1 - \alpha_{t-1}}\boldsymbol{\epsilon_\theta}(\boldsymbol{x}_t, t), \tag{1}$$

where $\boldsymbol{\epsilon_\theta}(\boldsymbol{x}_t, t)$ is parameterized by a neural network and trained to predict the noise $\boldsymbol{\epsilon}_t$ at time $t$. From previous works Zhang et al. (2024b); Luo (2022), the first term in Equation (1), defined as $\boldsymbol{f_{\theta,t}}(\boldsymbol{x}_t)$, is the *posterior mean predictor* (PMP) that predict the posterior mean $\mathbb{E}[\boldsymbol{x}_0|\boldsymbol{x}_t]$. DDIM could also be applied to a clean sample $\boldsymbol{x}_0$ and generate the corresponding noisy $\boldsymbol{x}_t$ at time $t$, named DDIM Inversion. One sampling step of DDIM inversion is similar to Equation (1), by mapping from $\boldsymbol{x}_{t-1}$ to $\boldsymbol{x}_t$. For any $t_1$ and $t_2$ with $t_2 > t_1$, we denote multi-time steps DDIM operator and its inversion as $\boldsymbol{x}_{t_1} = \text{DDIM}(\boldsymbol{x}_{t_2}, t_1)$ and $\boldsymbol{x}_{t_2} = \text{DDIM-Inv}(\boldsymbol{x}_{t_1}, t_2)$.

**Text-to-image (T2I) diffusion models & classifier-free guidance (CFG).**   The diffusion model can be generalized from unconditional to T2I (Rombach et al., 2022; Esser et al., 2024), where the latter enables controllable image generation $\boldsymbol{x}_0$ guided by a text prompt $\boldsymbol{c}$. In more detail, when training T2I diffusion models, we optimize a conditional denoising function $\boldsymbol{\epsilon_\theta}(\boldsymbol{x}_t, t, \boldsymbol{c})$. For sampling, we employ a technique called *classifier-free guidance* (CFG) (Ho & Salimans, 2022), which substitutes the unconditional denoiser $\boldsymbol{\epsilon_\theta}(\boldsymbol{x}_t, t)$ in Equation (1) with its conditional counterpart $\tilde{\boldsymbol{\epsilon}}_\theta(\boldsymbol{x}_t, t, \boldsymbol{c})$ that can be described as $\tilde{\boldsymbol{\epsilon}}_\theta(\boldsymbol{x}_t, t, \boldsymbol{c}) = (1 - \eta)\boldsymbol{\epsilon_\theta}(\boldsymbol{x}_t, t, \varnothing) + \eta\boldsymbol{\epsilon_\theta}(\boldsymbol{x}_t, t, \boldsymbol{c})$. Here, $\varnothing$ denotes the empty prompt and $\eta > 0$ denotes the strength for the classifier-free guidance. For simplification, for any $t_1$ and $t_2$ with $t_2 > t_1$, we denote multi-time steps CFG operator as $\boldsymbol{x}_{t_1} = \text{CFG}(\boldsymbol{x}_{t_2}, t_1, \boldsymbol{c})$. DDIM and DDIM inversion could also be generalized to T2I version, denotes as $\boldsymbol{x}_{t_1} = \text{DDIM}(\boldsymbol{x}_{t_2}, t_1, \boldsymbol{c})$ and $\boldsymbol{x}_{t_2} = \text{DDIM-Inv}(\boldsymbol{x}_{t_1}, t_2, \boldsymbol{c})$.

### 2.2 LOCAL LINEARITY AND INTRINSIC LOW-DIMENSIONALITY IN PMP

In this work, we will leverage two key properties of the PMP $\boldsymbol{f_{\theta,t}}(\boldsymbol{x}_t)$ introduced in Equation (1) for watermarking diffusion models. Parts of these properties have been previously identified in recent papers (Wang et al., 2024; Manor & Michaeli, 2024b;a), and they have been extensively studied in (Chen et al., 2024). At one given timestep $t \in [0, T]$, let us consider the first-order Taylor expansion of the PMP $\boldsymbol{f_{\theta,t}}(\boldsymbol{x}_t + \lambda\Delta\boldsymbol{x})$ at the point $\boldsymbol{x}_t$:

$$\boxed{\boldsymbol{l_\theta}(\boldsymbol{x}_t; \lambda\Delta\boldsymbol{x}) := \boldsymbol{f_{\theta,t}}(\boldsymbol{x}_t) + \lambda\boldsymbol{J_{\theta,t}}(\boldsymbol{x}_t) \cdot \Delta\boldsymbol{x},} \tag{2}$$

where $\Delta\boldsymbol{x} \in \mathbb{S}^{d-1}$ is a perturbation direction with unit length, $\lambda \in \mathbb{R}$ is the perturbation strength, and $\boldsymbol{J_{\theta,t}}(\boldsymbol{x}_t) = \nabla_{\boldsymbol{x}_t}\boldsymbol{f_{\theta,t}}(\boldsymbol{x}_t)$ is the Jacobian of $\boldsymbol{f_{\theta,t}}(\boldsymbol{x}_t)$. As shown in (Chen et al., 2024), it has

**Algorithm 1** Unconditional Shallow Diffuse

---

1: **Inject watermark:**
2: **Input**: original image $\boldsymbol{x}_0$ for the user scenario (initial random seed $\boldsymbol{x}_T$ for the server scenario), watermark $\lambda \Delta \boldsymbol{x}$, embedding timestep $t$,
3: **Output**: watermarked image $\boldsymbol{x}_0^{*\mathcal{W}}$,
4: **if** user scenario **then**
5:     $\boldsymbol{x}_t = \texttt{DDIM-Inv}\,(\boldsymbol{x}_0, t)$
6: **else** server scenario
7:     $\boldsymbol{x}_t = \texttt{DDIM}\,(\boldsymbol{x}_T, t)$
8: **end if**
9: $\boldsymbol{x}_t^{\mathcal{W}} \leftarrow \boldsymbol{x}_t + \lambda \Delta \boldsymbol{x}, \boldsymbol{x}_0^{\mathcal{W}} \leftarrow \texttt{DDIM}\left(\boldsymbol{x}_t^{\mathcal{W}}, 0\right)$                       ▷ Embed watermark
10: $\boldsymbol{x}_0^* \leftarrow \texttt{DDIM}\,(\boldsymbol{x}_t, 0), \boldsymbol{x}_0^{*\mathcal{W}} \leftarrow \texttt{ChannelAverage}\left(\boldsymbol{x}_0^{\mathcal{W}}, \boldsymbol{x}_0^*\right)$         ▷ Channel Average
11: **Return:** $\boldsymbol{x}_0^{*\mathcal{W}}$
12:
13: **Detect watermark:**
14: **Input**: Attacked image $\bar{\boldsymbol{x}}_0^{\mathcal{W}}$, watermark $\lambda \Delta \boldsymbol{x}$, embedding timestep $t$,
15: **Output**: Distance score $\eta$,
16: $\bar{\boldsymbol{x}}_t^{\mathcal{W}} \leftarrow \texttt{DDIM-Inv}\left(\bar{\boldsymbol{x}}_0^{\mathcal{W}}, t\right)$
17: $\eta = \texttt{Detector}\left(\bar{\boldsymbol{x}}_t^{\mathcal{W}}, \lambda \Delta \boldsymbol{x}\right)$
18: **Return:** $\eta$

---

been found that within a certain range of noise levels, the learned PMP $\boldsymbol{f}_{\boldsymbol{\theta},t}$ exhibits local linearity, and its Jacobian $\boldsymbol{J}_{\boldsymbol{\theta},t} \in \mathbb{R}^{d \times d}$ is low rank:

- **Low-rankness of the Jacobian $\boldsymbol{J}_{\boldsymbol{\theta},t}(\boldsymbol{x}_t)$.** As shown in Figure 2(a) of (Chen et al., 2024), the *rank ratio* for $t \in [0, T]$ *consistently* displays a U-shaped pattern across various network architectures and datasets: (*i*) it is close to 1 near either the pure noise $t = T$ or the clean image $t = 0$, (*ii*) $\boldsymbol{J}_{\boldsymbol{\theta},t}(\boldsymbol{x}_t)$ is low-rank (i.e., the numerical rank ratio less than $10^{-2}$) for all diffusion models within the range $t \in [0.2T, 0.7T]$, (*iii*) it achieves the lowest value around mid-to-late timestep, slightly differs on different architectures and datasets.

- **Local linearity of the PMP $\boldsymbol{f}_{\boldsymbol{\theta},t}(\boldsymbol{x}_t)$.** As shown in Figure 2(b) of (Chen et al., 2024), the mapping $\boldsymbol{f}_{\boldsymbol{\theta},t}(\boldsymbol{x}_t)$ exhibits strong linearity across a large portion of the timesteps, which is consistently true among different architectures trained on different datasets. In particular, the work (Chen et al., 2024) evaluated the linearity of $\boldsymbol{f}_{\boldsymbol{\theta},t}(\boldsymbol{x}_t)$ at $t = 0.7T$ where the rank ratio is close to the lowest value, showing that $\boldsymbol{f}_{\boldsymbol{\theta},t}(\boldsymbol{x}_t + \lambda \Delta \boldsymbol{x}) \approx \boldsymbol{l}_{\boldsymbol{\theta}}(\boldsymbol{x}_t; \lambda \Delta \boldsymbol{x})$ even when $\lambda = 40$,

## 3   Watermarking by Shallow-Diffuse

In this section, we introduce Shallow Diffuse for watermarking diffusion models. Building on the benign properties of PMP discussed in Section 2.2, we explain how to inject and detect invisible watermarks in *unconditional* diffusion models in Section 3.1 and Section 3.2, respectively. Algorithm 1 outlines the overall watermarking method for unconditional diffusion models. In Section 3.3, we extend this approach to *text-to-image* diffusion models, illustrated in Figure 3.

### 3.1   Injecting invisible watermarks

Consider an unconditional diffusion model $\boldsymbol{\epsilon}_{\boldsymbol{\theta}}(\boldsymbol{x}_t, t)$ as we introduced in Section 2.1. Instead of injecting the watermark $\Delta \boldsymbol{x}$ in the initial noise, we inject it in a particular timestep $t \in [0, T]$ with

$$\boldsymbol{x}_t^{\mathcal{W}} = \boldsymbol{x}_t + \lambda \Delta \boldsymbol{x}, \tag{3}$$

where $\lambda \in \mathbb{R}$ is the watermarking strength, $\boldsymbol{x}_t = \texttt{DDIM-Inv}\,(\boldsymbol{x}_0, t)$ under the user scenario and $\boldsymbol{x}_t = \texttt{DDIM}\,(\boldsymbol{x}_T, t)$ under the server scenario. Based upon Section 2.2, we choose the timestep $t$ so that the Jacobian of the PMP $\boldsymbol{J}_{\boldsymbol{\theta},t}(\boldsymbol{x}_t) = \nabla_{\boldsymbol{x}_t} \boldsymbol{f}_{\boldsymbol{\theta},t}(\boldsymbol{x}_t)$ is *low-rank*. Moreover, based upon the linearity of PMP discussed in Section 2.2, we approximately have

$$\boldsymbol{f}_{\boldsymbol{\theta},t}(\boldsymbol{x}_t^{\mathcal{W}}) = \boldsymbol{f}_{\boldsymbol{\theta},t}(\boldsymbol{x}_t) + \lambda \underbrace{\boldsymbol{J}_{\boldsymbol{\theta},t}(\boldsymbol{x}_t)}_{\approx \boldsymbol{0}} \cdot \Delta \boldsymbol{x} \approx \boldsymbol{f}_{\boldsymbol{\theta},t}(\boldsymbol{x}_t) = \hat{\boldsymbol{x}}_{0,t}, \tag{4}$$

where we select the watermark $\Delta \boldsymbol{x}$ to span the entire space $\mathbb{R}^d$ *uniformly*; a more detailed discussion on the pattern design of $\Delta \boldsymbol{x}$ is provided in Section 3.2. The key intuition for Equation (4) to hold is that, when $r_t = \text{rank}(\boldsymbol{J}_{\boldsymbol{\theta},t}(\boldsymbol{x}_t)) \ll d$ is low, a significant proportion of $\lambda \Delta \boldsymbol{x}$ lies in the *null space* of $\boldsymbol{J}_{\boldsymbol{\theta},t}(\boldsymbol{x}_t)$ so that $\boldsymbol{J}_{\boldsymbol{\theta},t}(\boldsymbol{x}_t)\Delta \boldsymbol{x} \approx \boldsymbol{0}$.

Therefore, the selection of $t$ is based on ensuring that $\boldsymbol{f}_{\boldsymbol{\theta},t}(\boldsymbol{x}_t)$ is locally linear and that the dimensionality of its Jacobian $r_t \ll d$. In practice, we choose $t = 0.3T$ based on results from the ablation study in Section 4.3. As a results, the injection in Equation (4) maintains better consistency without changing the predicted $\boldsymbol{x}_0$. In the meanwhile, it is very robust because any attack on $\boldsymbol{x}_0$ would remain disentangled from the watermark, so that $\lambda \Delta \boldsymbol{x}$ remains detectable.

Although in practice we employ the DDIM method instead of PMP for sampling high-quality images, the above intuition still carries over to DDIM. From Equation (1), one step sampling of DDIM in terms of $\boldsymbol{f}_{\boldsymbol{\theta},t}(\boldsymbol{x}_t)$ becomes:

$$\boldsymbol{x}_{t-1} = \sqrt{\alpha_{t-1}} \underbrace{\boldsymbol{f}_{\boldsymbol{\theta},t}(\boldsymbol{x}_t)}_{\text{"predicted } \boldsymbol{x}_0\text{"}} + \frac{\sqrt{1-\alpha_{t-1}}}{\sqrt{1-\alpha_t}} \underbrace{(\boldsymbol{x}_t - \sqrt{\alpha_t}\boldsymbol{f}_{\boldsymbol{\theta},t}(\boldsymbol{x}_t))}_{\text{"the direction pointing to } \boldsymbol{x}_t\text{"}} . \tag{5}$$

As explained in Song et al. (2021a), the first term predicts $\boldsymbol{x}_0$ while the second term points towards $\boldsymbol{x}_t$. When we inject the watermark $\Delta \boldsymbol{x}$ into $\boldsymbol{x}_t$ as given in Equation (3), we know that

$$\boldsymbol{x}_{t-1}^{\mathcal{W}} = \sqrt{\alpha_{t-1}}\boldsymbol{f}_{\boldsymbol{\theta},t}(\boldsymbol{x}_t^{\mathcal{W}}) + \frac{\sqrt{1-\alpha_{t-1}}}{\sqrt{1-\alpha_t}}\left(\boldsymbol{x}_t^{\mathcal{W}} - \sqrt{\alpha_t}\boldsymbol{f}_{\boldsymbol{\theta},t}(\boldsymbol{x}_t^{\mathcal{W}})\right)$$

$$\approx \sqrt{\alpha_{t-1}}\boldsymbol{f}_{\boldsymbol{\theta},t}(\boldsymbol{x}_t) + \frac{\sqrt{1-\alpha_{t-1}}}{\sqrt{1-\alpha_t}}\left(\boldsymbol{x}_t + \textcolor{blue}{\lambda\Delta \boldsymbol{x}} - \sqrt{\alpha_t}\boldsymbol{f}_{\boldsymbol{\theta},t}(\boldsymbol{x}_t)\right), \tag{6}$$

where the second approximation follows from Equation (4). This implies that the watermark $\lambda\Delta \boldsymbol{x}$ is embedded into the DDIM sampling process entirely through the second term of Equation (6) and it decouples from the first which predicts $\boldsymbol{x}_0$. Therefore, similar to our analysis for PMP, the first term in equation 6 maintains the consistency of data generation, while the difference in second term highlighted by blue would be useful for detecting the watermark which we will discuss next. In Appendix D, we provide more rigorous proofs validating the consistency and detectability of our approach.

## 3.2 WATERMARK DESIGN AND DETECTION

Second, building on the watermark injection method described in Section 3.1, we discuss the design of the watermark pattern and the techniques for effective detection.

**Watermark pattern design.** Building on the method proposed by Wen et al. (2023a), we inject the watermark in the frequency domain to enhance robustness against adversarial attacks. Specifically, we adapt this approach by defining a watermark $\lambda\Delta \boldsymbol{x}$ for the input $\boldsymbol{x}_t$ at timestep $t$ as follows:

$$\lambda\Delta \boldsymbol{x} := \texttt{DFT-Inv}\left(\texttt{DFT}\left(\boldsymbol{x}_t\right) \odot (1 - \boldsymbol{M}) + \boldsymbol{W} \odot \boldsymbol{M}\right) - \boldsymbol{x}_t, \tag{7}$$

where the Hadamard product $\odot$ denotes the element-wise multiplication. Additionally, we have the following for Equation (7):

- **Transformation into the frequency domain.** Let $\texttt{DFT}(\cdot)$ and $\texttt{DFT-Inv}(\cdot)$ represent the forward and inverse Discrete Fourier Transform (DFT) operators, respectively. As shown in Equation (7), we first apply $\texttt{DFT}(\cdot)$ to transform $\boldsymbol{x}_t$ into the frequency domain, where we then introduce the watermark via a mask. Finally, the modified input is transformed back into the pixel domain using $\texttt{DFT-Inv}(\cdot)$.

- **The mask and key of watermarks.** $\boldsymbol{M}$ is the mask used to apply the watermark in the frequency domain as shown in the top-left of Figure 2, and $\boldsymbol{W}$ denotes the key of the watermark. Typically, the mask M is circular, with the white area representing 1 and the black area representing 0 in Figure 2, where we use it to modify specific frequency bands of the image. In the following, we discuss the design of $\boldsymbol{M}$ and $\boldsymbol{W}$ in detail.

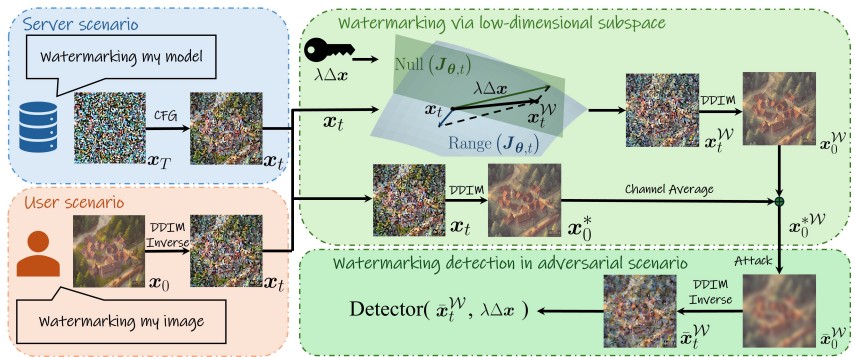

Figure 3: **Overview of Shallow Diffuse for T2I diffusion models.**

Previous methods (Wen et al., 2023a; Ci et al., 2024) design the mask $M$ to modify the low-frequency components of the initial noise input. While this approach works, as most of the energy in natural images is concentrated in the low-frequency range, it tends to distort the image when such watermarks are injected (see Figure 1 for an illustration). In contrast, as shown in Figure 2, we design the mask $M$ to target the high-frequency components of the image. Since high-frequency components capture fine details where the energy is less concentrated on these bands, modifying them results in less distortion of the original image. This is especially true in our case because we are modifying $x_t$, which is closer to $x_0$, compared to the initial noise used in (Wen et al., 2023a; Ci et al., 2024).To modify the high-frequency components, we apply the DFT without shifting and centering the zero frequency, as illustrated in the bottom-left of Figure 2.

In terms of designing the key $W$, we follow Wen et al. (2023a). The key $W$ is composed of multi-rings and each ring has the same value that is drawn from Gaussian distribution; see the top-right of Figure 2 for an illustration. Further ablation studies on the choice of $M$, $W$, and the effects of selecting low-frequency or high-frequency regions for watermarking can be found in Table 3.

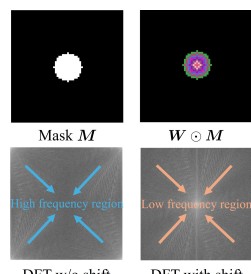

Figure 2: **Illustration of watermark patterns**.

**Watermark detection.** During watermark detection, suppose we are given a watermarked image $\bar{x}_0^{\mathcal{W}}$ with certain corruptions, we apply the DDIM Inversion to recover the watermarked image at timestep $t$, denoted as $\bar{x}_t^{\mathcal{W}} = \text{DDIM-Inv}\left(\bar{x}_0^{\mathcal{W}}, t\right)$. To detect the watermark, following Wen et al. (2023a); Zhang et al. (2024c), the $\text{Detector}(\cdot)$ in Algorithm 1 calculates the following p-value:

$$\eta = \frac{\text{sum}(M) \cdot \|M \odot W - M \odot \text{DFT}\left(\bar{x}_t^{\mathcal{W}}\right)\|_F^2}{\|M \odot \text{DFT}\left(\bar{x}_t^{\mathcal{W}}\right)\|_F^2}, \tag{8}$$

where $\text{sum}(\cdot)$ is the summation of all elements of the matrix. Ideally, if $\bar{x}_t^{\mathcal{W}}$ is a watermarked image, $M \odot W = M \odot \text{DFT}\left(\bar{x}_t^{\mathcal{W}}\right)$ and $\eta = 0$. When $\bar{x}_t^{\mathcal{W}}$ is a non-watermarked image, $M \odot W \neq M \odot \text{DFT}\left(\bar{x}_t^{\mathcal{W}}\right)$ and $\eta > 0$. By choosing a threshold $\eta_0$, non-watermarked images will have $\eta > \eta_0$ and watermarked images will have $\eta < \eta_0$. Theoretically, the derivation of the p-value $\eta$ could be found in Zhang et al. (2024c).

## 3.3 EXTENSION TO TEXT-TO-IMAGE (T2I) DIFFUSION MODELS

Up to this point, our discussion has focused exclusively on unconditional diffusion models. Next, we demonstrate how our approach can be readily extended to text-to-image (T2I) diffusion models, which are predominantly used in practice.

Figure 3 provides an overview of our method for T2I diffusion models, which can be flexibly applied to both server and user scenarios. Specifically,

Table 1: **Comparison under the server scenario.**

| Method | CLIP-Score ↑ | FID ↓ | Watermarking Robustness (AUC ↑/TPR@1%FPR↑) | | | | | |
|---|---|---|---|---|---|---|---|---|
| | | | Clean | JPEG | G.Blur | G.Noise | Color Jitter | Average |
| **Non-diffusion Method** | | | | | | | | |
| DwtDct | **0.3298** | 25.73 | 0.97/0.85 | 0.64/0.00 | 0.78/0.00 | 0.44/0.02 | 0.53/0.09 | 0.60/0.03 |
| DwtDctSvd | 0.3291 | 26.00 | **1.00/1.00** | 0.80/0.08 | 0.99/0.80 | 0.97/0.84 | 0.50/0.09 | 0.82/0.45 |
| RivaGAN | 0.3252 | **24.60** | 1.00/0.99 | **0.98/0.76** | **0.97/0.72** | **1.00/0.99** | **0.96/0.77** | **0.98/0.81** |
| **Diffusion Method** | | | | | | | | |
| Stable Diffusion w/o WM | 0.3286 | 25.56 | - | - | - | - | - | - |
| Stable Signature | 0.3622 | 30.86 | 1.00/1.00 | 0.99/0.76 | 0.57/0.00 | 0.71/0.14 | 0.96/0.87 | 0.81/0.46 |
| Tree-Ring Watermarks | 0.3310 | 25.82 | 1.00/1.00 | 0.99/0.97 | 0.98/0.98 | 0.94/0.50 | 0.96/0.67 | 0.97/0.80 |
| RingID | 0.3285 | 27.13 | 1.00/1.00 | 1.00/1.00 | 1.00/1.00 | 1.00/1.00 | 1.00/0.99 | 1.00/0.99 |
| Gaussian Shading | **0.3631** | 26.17 | 1.00/1.00 | 1.00/1.00 | 1.00/1.00 | 1.00/1.00 | 1.00/1.00 | 1.00/1.00 |
| **Shallow Diffuse (ours)** | 0.3285 | **25.58** | **1.00/1.00** | **1.00/1.00** | **1.00/1.00** | **1.00/1.00** | **1.00/1.00** | **1.00/1.00** |

- **Watermark injection.** Shallow Diffuse embeds watermarks into the noise corrupted image $x_t$ at a specific timestep $t = 0.3T$. In the **server scenario**, given $x_T \sim \mathcal{N}(0, I_d)$ and prompt $c$, we calculate $x_t = \text{CFG}(x_T, t, c)$. In the **user scenario**, given the generated image $x_0$, we compute $x_t = \text{DDIM-Inv}(x_0, t, \varnothing)$, using an empty prompt $\varnothing$. Next, similar to Section 3.1, we apply DDIM to obtain the watermarked image $x_0^{\mathcal{W}} = \text{DDIM}(x_t^{\mathcal{W}}, 0, \varnothing)$ and channel averaging $x_0^{*\mathcal{W}} \leftarrow \text{ChannelAverage}(x_0^{\mathcal{W}}, \text{DDIM}(x_t, 0))$. The detailed discussion about channel averaging is in Appendix B.

- **Watermark detection.** During watermark detection, suppose we are given a watermarked image $\bar{x}_0^{\mathcal{W}}$ with certain corruptions, we apply the DDIM Inversion to recover the watermarked image at timestep $t$, denoted as $\bar{x}_t^{\mathcal{W}} = \text{DDIM-Inv}(\bar{x}_0^{\mathcal{W}}, t, \varnothing)$. We detect the watermark $\Delta x$ in $\bar{x}_t^{\mathcal{W}}$ by calculating $\eta$ in Equation (8), with detail explained in Section 3.2.

## 4 EXPERIMENTS

In this section, we present a comprehensive set of experiments to demonstrate the robustness and consistency of *Shallow-Diffuse* across various datasets. Detailed experiment settings could be found in Appendix C.1. We begin by highlighting its performance in terms of robustness and consistency in both the server scenario (Section 4.1) and the user scenario (Section 4.2). Additionally, we compare Shallow Diffuse with other related works in the trade-off between robustness and consistency, as detailed in Appendix C.3. Moreover, we investigate the effect of timestep $t$ on both robustness and consistency, with results presented in Section 4.3. Lastly, we provide an ablation study on watermark pattern design, and channel averaging in Appendix C.

### 4.1 CONSISTENCY AND ROBUSTNESS UNDER THE SERVER SCENARIO

Table 1 compares the performance of Shallow Diffuse with other methods in the user scenario. For reference, we also apply stable diffusion to generate images from the same random seeds, without adding watermarks (referred to as "Stable Diffusion w/o WM" in Table 1). In terms of generation quality, Shallow Diffuse achieves the best FID score among the diffusion-based methods. Additionally, the FID and CLIP scores of Shallow Diffuse are very close to those of Stable Diffusion w/o WM. This similarity arises because the watermarked distribution produced by Shallow Diffuse remains highly consistent with the original generation distribution. Regarding robustness, Shallow Diffuse outperforms all other methods. Although both Gaussian Shading and RingID exhibit comparable generation quality and robustness in the server scenario, they are less suitable for the user scenario. Specifically, Gaussian Shading embeds the watermark into $x_T$, which is not accessible to the user, while RingID suffers from poor consistency, as demonstrated in Figure 1 and Table 2.

### 4.2 CONSISTENCY AND ROBUSTNESS UNDER THE USER SCENARIO

Table 2 presents a comparison of Shallow Diffuse's performance against other methods in the user scenario. In terms of consistency, Shallow Diffuse outperforms all other diffusion-based approaches. To measure the upper bound of diffusion-based methods, we apply stable diffusion with $\hat{x}_0 = \text{DDIM}(\text{DDIM-Inv}(x_0, t, \varnothing), 0, \varnothing)$, and measure the data consistency between $\hat{x}_0$ and $x_0$ (denotes in Stable Diffusion w/o WM in Table 2). The upper bound is constrained by errors introduced through DDIM inversion, and Shallow Diffuse comes the closest to reaching this limit. For

Table 2: **Comparison under the user scenario.**

| Method | PSNR ↑ | SSIM ↑ | LPIPS ↓ | Watermarking Robustness (AUC ↑/TPR@1%FPR↑) | | | | | |
| --- | --- | --- | --- | --- | --- | --- | --- | --- | --- |
| | | | | Clean | JPEG | G.Blur | G.Noise | Color Jitter | Average |
| **COCO** | | | | | | | | | |
| DwtDct | 37.88 | 0.97 | **0.02** | 0.98/0.83 | 0.48/0.02 | 0.50/0.00 | 0.30/0.06 | 0.57/0.16 | 0.46/0.06 |
| DwtDctSvd | 38.06 | **0.98** | **0.02** | **1.00/1.00** | 0.70/0.26 | 0.98/0.83 | 0.93/0.55 | 0.54/0.14 | 0.79/0.45 |
| RivaGAN | **40.57** | **0.98** | 0.04 | **1.00/1.00** | **1.00/1.00** | **0.99/0.86** | **1.00/0.99** | **0.97/0.83** | **0.99/0.92** |
| Stable Diffusion w/o WM | 32.28 | 0.78 | 0.06 | - | - | - | - | - | - |
| Tree-Ring Watermarks | 28.22 | 0.51 | 0.41 | **1.00/1.00** | 0.99/0.87 | 0.99/0.86 | 1.00/1.00 | 0.88/0.49 | 0.97/0.81 |
| RingID | 28.22 | 0.38 | 0.61 | **1.00/1.00** | **1.00/1.00** | **1.00/1.00** | 0.98/0.86 | **1.00/0.99** | 0.99/0.96 |
| **Shallow Diffuse (ours)** | **32.11** | **0.77** | **0.06** | **1.00/1.00** | **1.00/1.00** | **1.00/1.00** | **1.00/1.00** | **1.00/0.99** | **1.00/1.00** |
| **DiffusionDB** | | | | | | | | | |
| DwtDct | 37.77 | 0.96 | **0.02** | 0.96/0.76 | 0.71/0.23 | 0.96/0.70 | 0.35/0.01 | 0.52/0.12 | 0.64/0.27 |
| DwtDctSvd | 37.84 | 0.97 | **0.02** | **1.00/1.00** | 0.71/0.23 | 0.53/0.00 | 0.93/0.59 | 0.50/0.09 | 0.72/0.23 |
| RivaGAN | **40.6** | **0.98** | 0.04 | **1.00/0.98** | **1.00/0.72** | **0.96/0.76** | **0.99/0.94** | **0.96/0.76** | **0.98/0.80** |
| Stable Diffusion w/o WM | 33.42 | 0.85 | 0.03 | - | - | - | - | - | - |
| Tree-Ring Watermarks | 28.3 | 0.62 | 0.29 | **1.00/1.00** | 0.99/0.68 | 0.94/0.62 | **1.00/1.00** | 0.84/0.15 | 0.94/0.61 |
| RingID | 27.9 | 0.21 | 0.77 | **1.00/1.00** | **1.00/1.00** | **1.00/1.00** | 0.98/0.86 | 1.00/0.98 | 0.99/0.96 |
| **Shallow Diffuse (ours)** | **33.07** | **0.84** | **0.04** | **1.00/1.00** | 1.00/0.99 | 1.00/0.99 | **1.00/1.00** | **1.00/1.00** | **1.00/0.99** |
| **WikiArt** | | | | | | | | | |
| DwtDct | 38.84 | 0.97 | 0.02 | 0.96/0.75 | 0.44/0.00 | 0.51/0.01 | 0.26/0.00 | 0.49/0.12 | 0.43/0.03 |
| DwtDctSvd | 39.14 | 0.98 | 0.02 | **1.00/1.00** | 0.69/0.13 | 0.97/0.76 | 0.97/0.72 | 0.50/0.15 | 0.78/0.44 |
| RivaGAN | **40.44** | **0.98** | 0.05 | **1.00/1.00** | 0.97/0.81 | 1.00/0.95 | **1.00/1.00** | 0.90/0.65 | 0.97/0.85 |
| Stable Diffusion w/o WM | 31.6 | 0.7 | 0.09 | - | - | - | - | - | - |
| Tree-Ring Watermarks | 28.24 | 0.53 | 0.34 | **1.00/1.00** | 1.00/0.97 | 1.00/0.88 | **1.00/1.00** | 0.71/0.26 | 0.92/0.78 |
| RingID | 27.90 | 0.19 | 0.78 | **1.00/1.00** | **1.00/1.00** | **1.00/1.00** | 0.95/0.82 | 0.99/0.98 | 0.99/0.95 |
| **Shallow Diffuse (ours)** | **31.4** | **0.68** | **0.10** | **1.00/1.00** | 1.00/0.99 | 1.00/0.99 | **1.00/1.00** | **1.00/0.99** | **1.00/0.99** |

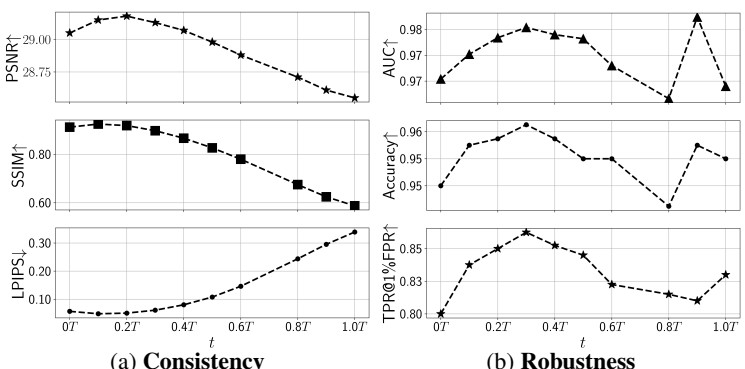

(a) **Consistency**        (b) **Robustness**

Figure 4: **Ablation study of the watermark at different timestep $t$.**

non-diffusion-based methods, which are not affected by DDIM inversion errors, better image consistency is achievable. As for the robustness, Shallow Diffuse outperforms all other methods in all three datasets. While RivaGAN achieves the best image consistency and comparable watermarking robustness to Shallow Diffuse in the user scenario, Shallow Diffuse is much more efficient. Unlike RivaGAN, which requires training for each individual image, Shallow Diffuse only involves the computational overhead of DDIM and DDIM inversion.

### 4.3 RELATION BETWEEN INJECTING TIMESTEP, CONSISTENCY AND ROBUSTNESS

Figure 4 shows the relationship between the watermark injection timestep $t$ and both consistency and robustness [1]. Shallow Diffuse achieves optimal consistency at $t = 0.2T$ and optimal robustness at $t = 0.3T$. In practice, we select $t = 0.3T$. This result aligns with the intuitive idea proposed in Section 3.1 and the theoretical analysis in Appendix D: low-dimensionality enhances both data generation consistency and watermark detection robustness. However, according to Chen et al. (2024), the optimal timestep $r_t$ for minimizing $r_t$ satisfies $t^* \in [0.5T, 0.7T]$. We believe the best consistency and robustness are not achieved at $t^*$ due to the error introduced by DDIM-Inv. As $t$ increases, this error grows, leading to a decline in both consistency and robustness. Therefore, the best tradeoff is reached at $t \in [0.2T, 0.3T]$, where $\boldsymbol{J}_{\boldsymbol{\theta},t}(\boldsymbol{x}_t)$ remains low-rank but $t$ is still below $t^*$.

---

[1]In this experiment, we do not incorporate additional techniques like channel averaging or enhanced watermark patterns. Therefore, when $t = 1.0T$, the method is equivalent to Tree-Ring Watermarks.

## 5   CONCLUSION

We proposed Shallow Diffuse, a novel and flexible watermarking technique that operates seamlessly in both server-side and user-side scenarios. By decoupling the watermark from the sampling process, Shallow Diffuse achieves enhanced robustness and greater consistency. Our theoretical analysis demonstrates both the consistency and detectability of the watermarks. Extensive experiments further validate the superiority of Shallow Diffuse over existing approaches.

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

## A RELATED WORK

### A.1 IMAGE WATERMARKING

Image watermarking has long been a crucial method for protecting intellectual property in computer vision (Cox et al., 2007; Solachidis & Pitas, 2001; Chang et al., 2005; Liu et al., 2019). Traditional techniques primarily focus on user-side watermarking, where watermarks are embedded into images post-generation. These methods (Al-Haj, 2007; Navas et al., 2008) typically operate in the frequency domain to ensure the watermarks are imperceptible. However, such watermarks remain vulnerable to adversarial attacks and can become undetectable after applying simple image manipulations like blurring.

Early deep learning-based approaches to watermarking (Zhang et al., 2024c; Fernandez et al., 2023; Ahmadi et al., 2020; Lee et al., 2020; Zhu et al., 2018) leveraged neural networks to embed watermarks. While these methods improved robustness and imperceptibility, they often suffer from high

computational costs during fine-tuning and lack flexibility. Each new watermark requires additional fine-tuning or retraining, limiting their practicality.

More recently, diffusion model-based watermarking techniques have gained attraction due to their ability to seamlessly integrate watermarks during the generative process without incurring extra computational costs. Techniques such as Wen et al. (2023a); Yang et al. (2024); Ci et al. (2024) embed watermarks directly into the initial noise and retrieve the watermark by reversing the diffusion process. These methods enhance robustness and invisibility but are typically restricted to server-side watermarking, requiring access to the initial random seed. Moreover, the watermarks introduced by Wen et al. (2023a); Ci et al. (2024) significantly alter the data distribution, leading to variance towards watermarks in generated outputs (as shown in Figure 1).

In contrast to Wen et al. (2023a); Ci et al. (2024), our proposed shallow diffuse disentangles the watermark embedding from the generation process by leveraging the high-dimensional null space. This approach, both empirically and theoretically validated, significantly improves watermark consistency and robustness. To the best of our knowledge, this is the first method that supports watermark embedding for both server-side and user-side applications while maintaining high robustness and consistency.

### A.2 LOW-DIMENSIONAL SUBSPACE IN DIFFUSION MODEL

In recent years, there has been growing interest in understanding deep generative models through the lens of the manifold hypothesis (Loaiza-Ganem et al., 2024). This hypothesis suggests that high-dimensional real-world data actually lies in latent manifolds with a low intrinsic dimension. Focusing on diffusion models, Stanczuk et al. (2024) empirically and theoretically shows that the approximated score function (the gradient of the log density of a noise-corrupted data distribution) in diffusion models is orthogonal to a low-dimensional subspace. Building on this, Wang et al. (2024); Chen et al. (2024) find that the estimated posterior mean from diffusion models lies within this low-dimensional space. Additionally, Chen et al. (2024) discovers strong local linearity within the space, suggesting that it can be locally approximated by a linear subspace. This observation motivates our Assumption 1, where we assume the estimated posterior mean lies in a low-dimensional subspace.

Building upon these findings, Stanczuk et al. (2024); Kamkari et al. (2024) introduce a local intrinsic dimension estimator, while Loaiza-Ganem et al. (2024) proposes a method for detecting out-of-domain data. Wang et al. (2024) offers theoretical insights into how diffusion model training transitions from memorization to generalization, and Chen et al. (2024); Manor & Michaeli (2024b) explores the semantic basis of the subspace to achieve disentangled image editing. Unlike these previous works, our approach leverages the low-dimensional subspace for watermarking, where both empirical and theoretical evidence demonstrates that this subspace enhances robustness and consistency.

## B CHANNEL AVERAGING

### B.1 TECHNIQUE DETAILS

Natural images have multiple channels denoted by $C$. Instead of applying watermark $\lambda\Delta$ to all channels of $\boldsymbol{x}_t$, we can apply the watermark to a specific channel $c$ to make it even more invisible and robust. For this consideration, let us reshape the image $\boldsymbol{x}_t$ and the watermark $\Delta\boldsymbol{x}$ into the form $\boldsymbol{x}_t \in \mathbb{R}^{H \times W \times C}, \lambda\Delta\boldsymbol{x} \in \mathbb{R}^{H \times W \times C}$, where $H$, $W$, and $C$ represent the height, width, and channel dimensions for the image, respectively. These dimensions satisfy $HWC = d$.

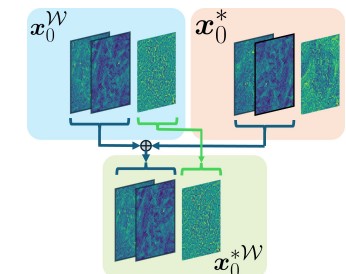

Figure 5: Illustration of channel average

Denote $[\boldsymbol{x}_t]_i \in \mathbb{R}^{H \times W}$ as the $i$th channel of $\boldsymbol{x}_t$, with $i \in [C]$. Thus $[\boldsymbol{x}_t^{\mathcal{W}}]_c = [\boldsymbol{x}_t]_c + [\lambda\Delta\boldsymbol{x}]_c$ and $[\boldsymbol{x}_t^{\mathcal{W}}]_i = [\boldsymbol{x}_t]_i$ for $i \neq c$. For the watermark in Equation (3), the channel averaging is defined

as:

$$[\boldsymbol{x}_0^{*\mathcal{W}}]_i = \texttt{ChannelAverage}\left(\boldsymbol{x}_0^{\mathcal{W}}, \boldsymbol{x}_0^*\right), \tag{9}$$

$$= \left\{ \begin{array}{l} [\boldsymbol{x}_0^{\mathcal{W}}]_i, i = c \\ (1-\gamma)[\boldsymbol{x}_0^{\mathcal{W}}]_i + \gamma[\boldsymbol{x}_0^*]_i, i \neq c \end{array} \right., \tag{10}$$

where we applied $\gamma = 1$. In our experiments, we found that we can increase both imperceptibility and robustness by further employing this simple approach. See our ablation study in Appendix C.5 for a more detailed analysis.

# C  ADDITIONAL EXPERIMENTS

## C.1  DETAILS ABOUT EXPERIMENT SETTINGS

**Baseline**  For the server scenario, we select the following methods as baselines: DWtDct Cox et al. (2007), DwtDctSvd Cox et al. (2007), RivaGAN Zhang et al. (2019), Stable Signature Fernandez et al. (2023), Tree-Ring Watermarks Wen et al. (2023a), RingId Ci et al. (2024), and Gaussian Shading Yang et al. (2024). In the user scenario, we adopt the same baseline methods, except for Stable Signature and Gaussian Shading, as these methods are not suitable for this setting.

**Datasets**  We use Stable Diffusion 2.1 (Rombach et al., 2022) as the underlying model for our experiments, applying Shallow diffusion within its latent space. For the server scenario (Section 4.1), all diffusion-based methods are based on the same Stable Diffusion, with the original images $\boldsymbol{x}_0$ generated from identical initial seeds $\boldsymbol{x}_T$. Non-diffusion methods are applied to these same original images $\boldsymbol{x}_0$ in a post-watermarking process. A total of 5000 original images are generated for evaluation in this scenario. For the user scenario (Section 4.2), we utilize the MS-COCO Lin et al. (2014), WikiArt Tan et al. (2019), and DiffusionDB datasets Wang et al. (2022). The first two are real-world datasets, while DiffusionDB is a collection of diffusion model-generated images. From each dataset, we select 500 images for evaluation. For the remaining experiments in Appendix C.3, Section 4.3, Appendix C, we use the server scenario and sample 100 images for evaluation.

**Metric**  To evaluate image consistency under the user scenario, we use peak signal-to-noise ratio (PSNR) Jähne (2005), structural similarity index measure (SSIM) Wang et al. (2004), and Learned Perceptual Image Patch Similarity (LPIPS) Zhang et al. (2018), comparing watermarked images to their original counterparts. In the server scenario, we assess the generation quality of the watermarked images using Contrastive Language-Image Pretraining Score (CLIP-Score) Radford et al. (2021) and Fréchet Inception Distance (FID) Heusel et al. (2017). To evaluate robustness, we vary the threshold $\eta_0$ and plot the true positive rate (TPR) against the false positive rate (FPR) for the receiver operating characteristic (ROC) curve. We use the area under the curve (AUC) and TPR when FPR = 0.01 (TPR @1% FPR) as robustness metrics. Robustness is evaluated both under clean conditions (no attacks) and with various attacks, including JPEG compression, Gaussian blurring (G.Blur), Gaussian noise, and color jitter. Details of these attacks are provided in Appendix C.2.

## C.2  DETAILS ABOUT ATTACKS

In this work, we intensively tested our method on four different watermarking attacks, both in the server scenario and in the user scenario. These watermarking attacks represent the most common image distortion methods in real life, including

- JPEG compression with a compression rate of 25%
- Gaussian blurring (G.Blur) with an $8 \times 8$ filter size
- Gaussian noise (G.Noise) with $\sigma = 0.1$
- Color jitter with brightness factor uniformly ranges between 0 and 6

## C.3  TRADE-OFF BETWEEN CONSISTENCY AND ROBUSTNESS

Figure 6 illustrates the trade-off between consistency and robustness for Shallow Diffuse and other baselines. As the radius of $M$ increases, the watermark intensity $\lambda$ also increases, reducing image

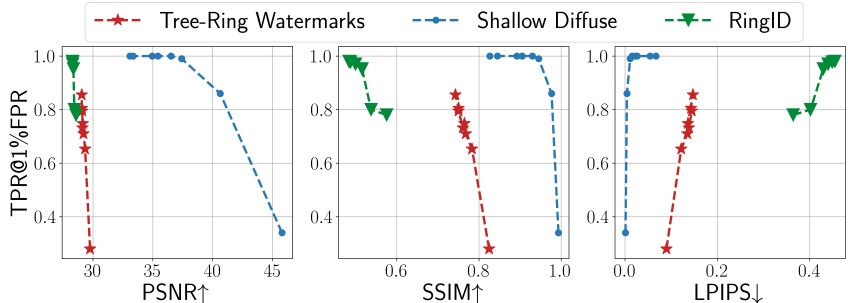

Figure 6: **Trade-off between consistency and robustness for Tree-Ring Watermarks, RingID, and Shallow Diffuse.**

Table 3: **Ablation study of different watermark patterns.**

| Method & Dataset | | | PSNR ↑ | SSIM ↑ | LPIPS ↓ | Average Watermarking Robustness (AUC ↑/TPR@1%FPR↑) |
|---|---|---|---|---|---|---|
| Frequency Region | Shape | Distribution | | | | |
| Low | Circle | Zero | 29.10 | 0.90 | 0.06 | 0.93/0.65 |
| Low | Circle | Rand | 29.37 | 0.92 | 0.05 | 0.92/0.25 |
| Low | Circle | Rotational Rand | 29.13 | 0.90 | 0.06 | 1.00/1.00 |
| Low | Ring | Zero | 36.20 | 0.95 | 0.02 | 0.78/0.35 |
| Low | Ring | Rand | 38.23 | 0.97 | 0.01 | 0.87/0.49 |
| Low | Ring | Rotational Rand | 35.23 | 0.93 | 0.02 | 0.99/0.98 |
| High | Circle | Zero | 38.3 | 0.96 | 0.01 | 0.80/0.34 |
| High | Circle | Rand | 42.3 | 0.98 | 0.004 | 0.86/0.35 |
| High | Circle | Rotational Rand | 38.0 | 0.94 | 0.01 | 1.00/1.00 |

consistency but improving robustness. By adjusting the radius of $M$, we plot the trade-off using PSNR, SSIM, and LPIPS against TPR@1%FPR. From Figure 6, curve of Shallow Diffuse is consistently above the curve of Tree-Ring Watermarks and RingID, demonstrating Shallow Diffuse's better consistency at the same level of robustness.

### C.4 ABLATION STUDY OF DIFFERENT WATERMARK PATTERNS

### C.5 ABLATION STUDY OF CHANNEL AVERAGE

### C.6 ABLATION STUDY OF WATERMARKING EMBEDDED CHANNEL.

## D THEORETICAL JUSTIFICATION

In this section, we provide theoretical justifications for the consistency and the detectability of Shallow Diffuse introduced in Section 3 for unconditional diffusion models. First, we make the following assumptions on the watermark and the diffusion model process.

**Assumption 1.** *Suppose the following hold for the PMP $\boldsymbol{f}_{\boldsymbol{\theta},t}(\boldsymbol{x}_t)$:*

- ***Linearity:*** *For any small $t$ and $\Delta\boldsymbol{x} \in \mathbb{S}^{d-1}$, we always have*

$$\boldsymbol{f}_{\boldsymbol{\theta},t}(\boldsymbol{x}_t + \lambda\Delta\boldsymbol{x}) = \boldsymbol{f}_{\boldsymbol{\theta},t}(\boldsymbol{x}_t) + \lambda\boldsymbol{J}_{\boldsymbol{\theta},t}(\boldsymbol{x}_t)\Delta\boldsymbol{x}.$$

- ***L-Lipschitz continuous:*** *we assume that $\boldsymbol{f}_{\boldsymbol{\theta},t}(\boldsymbol{x})$ is a $L$-Lipschiz continuous at every $t$:*

$$||\boldsymbol{J}_{\boldsymbol{\theta},t}(\boldsymbol{x})||_2 \leq L, \quad \forall\boldsymbol{x} \in \mathbb{R}^d.$$

It should be noted that our assumptions are mild. The $L$-Lipschitz continuity is a common assumption for analysis. The approximated linearity have been shown in (Chen et al., 2024) with the assumption of data distribution to be a mixture of low-rank Gaussians. Here, we assume the linearity to be exact for the ease of analysis, and it can be generalized to approximate linear case.

Now consider injecting a watermark $\lambda\Delta\boldsymbol{x}$ in Equation (3), where $\lambda > 0$ is a scaling factor and $\Delta\boldsymbol{x}$ is a *random* vector uniformly distributed on the unit hypersphere $\mathbb{S}^{d-1}$, i.e., $\Delta\boldsymbol{x} \sim \mathrm{U}(\mathbb{S}^{d-1})$. Then the following hold for the PMP $\boldsymbol{f}_{\boldsymbol{\theta},t}(\boldsymbol{x}_t)$.

Table 4: **Ablation study of channel average.**

| Channel average intensity $\gamma$ | PSNR ↑ | SSIM ↑ | LPIPS ↓ | Watermarking Robustness (TPR@1%FPR↑) | | | | |
|---|---|---|---|---|---|---|---|---|
| | | | | Clean | JPEG | G.Blur | G.Noise | Color Jitter |
| 0 | 37.1103 | 0.941 | 0.0154 | 1.0000 | 1.0000 | 0.9971 | 1.0000 | 0.9584 |
| 1.0 | 36.6352 | 0.931 | 0.0151 | 1.0000 | 1.0000 | 1.0000 | 1.0000 | 1.0000 |

Table 5: **Ablation study of watermarking embedded channel.**

| Watermark embedding channel | PSNR ↑ | SSIM ↑ | LPIPS ↓ | Watermarking Robustness (TPR@1%FPR↑) | | | | |
|---|---|---|---|---|---|---|---|---|
| | | | | Clean | JPEG | G.Blur | G.Noise | Color Jitter |
| 0th | 36.46 | 0.93 | 0.02 | 1.00 | 1.00 | 1.00 | 1.00 | 0.99 |
| 1th | 36.57 | 0.93 | 0.02 | 1.00 | 1.00 | 1.00 | 1.00 | 0.99 |
| 2th | 36.13 | 0.92 | 0.02 | 1.00 | 1.00 | 1.00 | 1.00 | 1.00 |
| 3th | 36.64 | 0.93 | 0.02 | 1.00 | 1.00 | 1.00 | 1.00 | 1.00 |
| 1th + 2th + 3th | 33.19 | 0.83 | 0.05 | 1.00 | 1.00 | 1.00 | 1.00 | 0.95 |

**Theorem 1** (Consistency of the watermarks). *Suppose Assumption 1 holds and $\Delta \boldsymbol{x} \sim \mathrm{U}(\mathbb{S}^{d-1})$. Let us define $\hat{\boldsymbol{x}}_{0,t}^{\mathcal{W}} \coloneqq \boldsymbol{f}_{\boldsymbol{\theta},t}(\boldsymbol{x}_t + \lambda \Delta \boldsymbol{x})$, $\hat{\boldsymbol{x}}_{0,t} \coloneqq \boldsymbol{f}_{\boldsymbol{\theta},t}(\boldsymbol{x}_t + \lambda \Delta \boldsymbol{x})$. The $\ell_2$-norm distance between $\hat{\boldsymbol{x}}_{0,t}^{\mathcal{W}}$ and $\hat{\boldsymbol{x}}_{0,t}$ can be bounded by:*

$$||\hat{\boldsymbol{x}}_{0,t}^{\mathcal{W}} - \hat{\boldsymbol{x}}_{0,t}||_2 \leq \lambda L h(r_t), \tag{11}$$

*with probability at least $1 - r_t^{-1}$. Here, $h(r_t) = \sqrt{\dfrac{r_t}{d} + \sqrt{\dfrac{18\pi^3}{d-2}\log(2r_t)}}$.*

Our Theorem 1 guarantees that adding the watermark $\lambda \Delta \boldsymbol{x}$ would only change the estimation by an amount of $\lambda L h(r_t)$ with a constant probability. In particular, when $r_t$ is small, it implies that the change in the prediction would be small. Given the relationship between PMP and DDIM in equation 1, the consistency also applies to the practical use. On the other hand, in the following we show that the injected watermark can be detected based upon the second term in Equation (6).

**Theorem 2** (Detectability of the watermarks). *Suppose Assumption 1 holds and $\Delta \boldsymbol{x} \sim \mathrm{U}(\mathbb{S}^{d-1})$. With $\boldsymbol{x}_t^{\mathcal{W}}$ given in Equation (3), define $\boldsymbol{x}_{t-1}^{\mathcal{W}} = \mathrm{DDIM}(\boldsymbol{x}_t^{\mathcal{W}}, t-1)$ and $\bar{\boldsymbol{x}}_t^{\mathcal{W}} = \mathrm{DDIM\text{-}Inv}(\tilde{\boldsymbol{x}}_{t-1}^{\mathcal{W}}, t)$. The $\ell_2$-norm distance between $\tilde{\boldsymbol{x}}_t^{\mathcal{W}}$ and $\boldsymbol{x}_t^{\mathcal{W}}$ can be bounded by:*

$$||\bar{\boldsymbol{x}}_t^{\mathcal{W}} - \boldsymbol{x}_t^{\mathcal{W}}||_2 \leq \lambda L \left(-g\left(\alpha_t, \alpha_{t-1}\right) + g\left(\alpha_{t-1}, \alpha_t\right)\left(1 - Lg\left(\alpha_t, \alpha_{t-1}\right)\right)\right) h(\max\{r_{t-1}, r_t\}) \tag{12}$$

*with probability at least $1 - r_t^{-1} - r_{t-1}^{-1}$. Here, $g(x, y) \coloneqq \dfrac{\sqrt{1-y}\sqrt{x} - \sqrt{1-x}\sqrt{y}}{\sqrt{1-x}}$, $\forall x, y \in (0, 1)$.*

Here $-g\left(\alpha_t, \alpha_{t-1}\right) + g\left(\alpha_{t-1}, \alpha_t\right)\left(1 - Lg\left(\alpha_t, \alpha_{t-1}\right)\right)$ is a small number under the $\alpha_t$ designed for variance preserving (VP) noise scheduler Ho et al. (2020) and $h(\max\{r_{t-1}, r_t\})$ is small when $r_t$ is small. This indicates that the difference between $\bar{\boldsymbol{x}}_t^{\mathcal{W}}$ and $\boldsymbol{x}_t^{\mathcal{W}}$ is small when $r_t$ is small and $\boldsymbol{x}_t^{\mathcal{W}}$ could be recovered by $\bar{\boldsymbol{x}}_t^{\mathcal{W}}$ from one-step DDIM. Therefore, Theorem 2 implies that the injected watermark can be detected with constant probability.

# E    PROOFS IN SECTION D

## E.1    PROOFS OF THEOREM 1

*Proof of Theorem 1.* According to Assumption 1, we have $||\hat{\boldsymbol{x}}_{0,t}^{\mathcal{W}} - \hat{\boldsymbol{x}}_{0,t}||_2^2 = \lambda ||\boldsymbol{J}_{\boldsymbol{\theta},t}(\boldsymbol{x}_t) \cdot \Delta \boldsymbol{x}||_2^2$. From Levy's Lemma proposed in Popescu et al. (2006), given function $||\boldsymbol{J}_{\boldsymbol{\theta},t}(\boldsymbol{x}_t) \cdot \Delta \boldsymbol{x}||_2^2 : \mathbb{S}^{d-1} \to \mathbb{R}$ we have:

$$\mathbb{P}\left(\left|||\boldsymbol{J}_{\boldsymbol{\theta},t}(\boldsymbol{x}_t) \cdot \Delta \boldsymbol{x}||_2^2 - \mathbb{E}\left[||\boldsymbol{J}_{\boldsymbol{\theta},t}(\boldsymbol{x}_t) \cdot \Delta \boldsymbol{x}||_2^2\right]\right| \geq \epsilon\right) \leq 2\exp\left(\frac{-C(d-2)\epsilon^2}{L^2}\right),$$

given $L$ to be the Lipschitz constant of $||\boldsymbol{J}_{\boldsymbol{\theta},t}(\boldsymbol{x}_t)||_2^2$ and $C$ is a positive constant (which can be taken to be $C = (18\pi^3)^{-1}$). From Lemma 2 and Lemma 3, we have:

$$\mathbb{P}\left(\left|||\boldsymbol{J}_{\boldsymbol{\theta},t}(\boldsymbol{x}_t)\cdot\Delta\boldsymbol{x}||_2^2 - \frac{||\boldsymbol{J}_{\boldsymbol{\theta},t}(\boldsymbol{x}_t)||_F^2}{d}\right| \geq \epsilon\right) \leq 2\exp\left(\frac{-(18\pi^3)^{-1}(d-2)\epsilon^2}{||\boldsymbol{J}_{\boldsymbol{\theta},t}(\boldsymbol{x}_t)||_2^4}\right).$$

Define $\dfrac{1}{r_t}$ as the desired probability level, set

$$\frac{1}{r_t} = 2\exp\left(\frac{-(18\pi^3)^{-1}(d-2)\epsilon^2}{||\boldsymbol{J}_{\boldsymbol{\theta},t}(\boldsymbol{x}_t)||_2^4}\right),$$

Solving for $\epsilon$:

$$\epsilon = ||\boldsymbol{J}_{\boldsymbol{\theta},t}(\boldsymbol{x}_t)||_2^2\sqrt{\frac{18\pi^3}{d-2}\log(2r_t)}.$$

Therefore, with probability $1 - \dfrac{1}{r_t}$, we have:

$$||\hat{\boldsymbol{x}}_{0,t}^{\mathcal{W}} - \hat{\boldsymbol{x}}_{0,t}||_2^2 = \lambda^2||\boldsymbol{J}_{\boldsymbol{\theta},t}(\boldsymbol{x}_t)\cdot\Delta\boldsymbol{x}||_2^2,$$

$$\leq \frac{\lambda^2||\boldsymbol{J}_{\boldsymbol{\theta},t}(\boldsymbol{x}_t)||_F^2}{d} + \lambda^2||\boldsymbol{J}_{\boldsymbol{\theta},t}(\boldsymbol{x}_t)||_2^2\sqrt{\frac{18\pi^3}{d-2}\log(2r_t)},$$

$$\leq \lambda^2||\boldsymbol{J}_{\boldsymbol{\theta},t}(\boldsymbol{x}_t)||_2^2\left(\frac{r_t}{d} + \sqrt{\frac{18\pi^3}{d-2}\log(2r_t)}\right),$$

$$= \lambda^2 L^2\left(\frac{r_t}{d} + \sqrt{\frac{18\pi^3}{d-2}\log(2r_t)}\right),$$

where the last inequality is obtained from $||\boldsymbol{J}_{\boldsymbol{\theta},t}(\boldsymbol{x}_t)||_F^2 \leq r_t||\boldsymbol{J}_{\boldsymbol{\theta},t}(\boldsymbol{x}_t)||_2^2$. Therefore, with probability $1 - \dfrac{1}{r_t}$,

$$||\hat{\boldsymbol{x}}_{0,t}^{\mathcal{W}} - \hat{\boldsymbol{x}}_{0,t}||_2 \leq \lambda L\sqrt{\frac{r_t}{d} + \sqrt{\frac{18\pi^3}{d-2}\log(2r_t)}} = \lambda L h(r_t).$$

$\square$

*Proof of Theorem 2.* According to Equation (1), one step of DDIM sampling at timestep $t$ could be represented by PMP $\boldsymbol{f}_{\boldsymbol{\theta},t}(\boldsymbol{x}_t)$ as:

$$\boldsymbol{x}_{t-1} = \sqrt{\alpha_{t-1}}\boldsymbol{f}_{\boldsymbol{\theta},t}(\boldsymbol{x}_t) + \sqrt{1-\alpha_{t-1}}\left(\frac{\boldsymbol{x}_t - \sqrt{\alpha_t}\boldsymbol{f}_{\boldsymbol{\theta},t}(\boldsymbol{x}_t)}{\sqrt{1-\alpha_t}}\right), \tag{13}$$

$$= \sqrt{\frac{1-\alpha_{t-1}}{1-\alpha_t}}\boldsymbol{x}_t + \frac{\sqrt{1-\alpha_t}\sqrt{\alpha_{t-1}} - \sqrt{1-\alpha_{t-1}}\sqrt{\alpha_t}}{\sqrt{1-\alpha_t}}\boldsymbol{f}_{\boldsymbol{\theta},t}(\boldsymbol{x}_t), \tag{14}$$

If we inject a watermark $\lambda\Delta\boldsymbol{x}$ to $\boldsymbol{x}_t$, so $x_t^{\mathcal{W}} = \boldsymbol{x}_t + \lambda\Delta\boldsymbol{x}$. To solve $x_{t-1}^{\mathcal{W}}$, we could plugging Equation (2) to Equation (14), we could obtain:

$$\boldsymbol{x}_{t-1}^{\mathcal{W}} = \sqrt{\frac{1-\alpha_{t-1}}{1-\alpha_t}}\boldsymbol{x}_t^{\mathcal{W}} + \frac{\sqrt{1-\alpha_t}\sqrt{\alpha_{t-1}} - \sqrt{1-\alpha_{t-1}}\sqrt{\alpha_t}}{\sqrt{1-\alpha_t}}\boldsymbol{f}_{\boldsymbol{\theta},t}(\boldsymbol{x}_t^{\mathcal{W}}), \tag{15}$$

$$= \boldsymbol{x}_{t-1} + \sqrt{\frac{1-\alpha_{t-1}}{1-\alpha_t}}\lambda\Delta\boldsymbol{x} + \frac{\sqrt{1-\alpha_t}\sqrt{\alpha_{t-1}} - \sqrt{1-\alpha_{t-1}}\sqrt{\alpha_t}}{\sqrt{1-\alpha_t}}\boldsymbol{J}_{\boldsymbol{\theta},t}(\boldsymbol{x}_t)\Delta\boldsymbol{x} \tag{16}$$

$$= \boldsymbol{x}_{t-1} + \lambda\underbrace{\left(\sqrt{\frac{1-\alpha_{t-1}}{1-\alpha_t}}\boldsymbol{I} + \frac{\sqrt{1-\alpha_t}\sqrt{\alpha_{t-1}} - \sqrt{1-\alpha_{t-1}}\sqrt{\alpha_t}}{\sqrt{1-\alpha_t}}\boldsymbol{J}_{\boldsymbol{\theta},t}(\boldsymbol{x}_t)\right)}_{:=\boldsymbol{W}_t}\Delta\boldsymbol{x}, \tag{17}$$

One step DDIM Inverse sampling at timestep $t-1$ could be represented by PMP $f_{\boldsymbol{\theta},t}(\boldsymbol{x}_t)$ as:

$$\boldsymbol{x}_t = \sqrt{\frac{1-\alpha_t}{1-\alpha_{t-1}}}\boldsymbol{x}_{t-1} + \frac{\sqrt{1-\alpha_{t-1}}\sqrt{\alpha_t}-\sqrt{1-\alpha_t}\sqrt{\alpha_{t-1}}}{\sqrt{1-\alpha_{t-1}}}\boldsymbol{f}_{\boldsymbol{\theta},t-1}(\boldsymbol{x}_{t-1}), \tag{18}$$

To detect the watermark, we apply one step DDIM Inverse on $\boldsymbol{x}_{t-1}^{\mathcal{W}}$ at timestep $t-1$ to obtain $\tilde{x}_t^{\mathcal{W}}$:

$$
\begin{aligned}
\tilde{x}_t^{\mathcal{W}} &= \sqrt{\frac{1-\alpha_t}{1-\alpha_{t-1}}}\boldsymbol{x}_{t-1}^{\mathcal{W}} + \frac{\sqrt{1-\alpha_{t-1}}\sqrt{\alpha_t}-\sqrt{1-\alpha_t}\sqrt{\alpha_{t-1}}}{\sqrt{1-\alpha_{t-1}}}\boldsymbol{f}_{\boldsymbol{\theta},t-1}(\boldsymbol{x}_{t-1}^{\mathcal{W}}), \\
&= \boldsymbol{x}_t + \lambda \underbrace{\left( \sqrt{\frac{1-\alpha_t}{1-\alpha_{t-1}}}\boldsymbol{I} + \frac{\sqrt{1-\alpha_{t-1}}\sqrt{\alpha_t}-\sqrt{1-\alpha_t}\sqrt{\alpha_{t-1}}}{\sqrt{1-\alpha_{t-1}}}\boldsymbol{J}_{\boldsymbol{\theta},t-1}(\boldsymbol{x}_{t-1}) \right)}_{:=\boldsymbol{W}_{t-1}} \boldsymbol{W}_t\Delta\boldsymbol{x}, \\
&= \boldsymbol{x}_t + \lambda\boldsymbol{W}_{t-1}\boldsymbol{W}_t\Delta\boldsymbol{x} = \boldsymbol{x}_t^{\mathcal{W}} + \lambda\left(\boldsymbol{W}_{t-1}\boldsymbol{W}_t - \boldsymbol{I}\right)\Delta\boldsymbol{x}.
\end{aligned}
$$

Therefore:

$$
\begin{aligned}
||\tilde{x}_t^{\mathcal{W}} - \boldsymbol{x}_t^{\mathcal{W}}||_2 &= \lambda||\left(\boldsymbol{W}_{t-1}\boldsymbol{W}_t - \boldsymbol{I}\right)\Delta\boldsymbol{x}||_2, \\
&= \lambda||\frac{\sqrt{1-\alpha_{t-1}}\sqrt{\alpha_t}-\sqrt{1-\alpha_t}\sqrt{\alpha_{t-1}}}{\sqrt{1-\alpha_t}}\boldsymbol{J}_{\boldsymbol{\theta},t-1}(\boldsymbol{x}_{t-1})\Delta\boldsymbol{x}, \\
&\quad + \frac{\sqrt{1-\alpha_t}\sqrt{\alpha_{t-1}}-\sqrt{1-\alpha_{t-1}}\sqrt{\alpha_t}}{\sqrt{1-\alpha_{t-1}}}\boldsymbol{J}_{\boldsymbol{\theta},t}(\boldsymbol{x}_t)\Delta\boldsymbol{x}, \\
&\quad - \frac{\left(\sqrt{1-\alpha_t}\sqrt{\alpha_{t-1}}-\sqrt{1-\alpha_{t-1}}\sqrt{\alpha_t}\right)^2}{\sqrt{1-\alpha_{t-1}}\sqrt{1-\alpha_t}}\boldsymbol{J}_{\boldsymbol{\theta},t-1}(\boldsymbol{x}_{t-1})\boldsymbol{J}_{\boldsymbol{\theta},t}(\boldsymbol{x}_t)\Delta\boldsymbol{x}||_2, \\
&\leq -\lambda g\left(\alpha_t,\alpha_{t-1}\right)||\boldsymbol{J}_{\boldsymbol{\theta},t-1}(\boldsymbol{x}_{t-1})\Delta\boldsymbol{x}||_2 + \lambda g\left(\alpha_{t-1},\alpha_t\right)||\boldsymbol{J}_{\boldsymbol{\theta},t}(\boldsymbol{x}_t)\Delta\boldsymbol{x}||_2 \\
&\quad - \lambda g\left(\alpha_{t-1},\alpha_t\right)g\left(\alpha_t,\alpha_{t-1}\right)||\boldsymbol{J}_{\boldsymbol{\theta},t-1}(\boldsymbol{x}_{t-1})\boldsymbol{J}_{\boldsymbol{\theta},t}(\boldsymbol{x}_t)\Delta\boldsymbol{x}||_2, \\
&\leq -\lambda g\left(\alpha_t,\alpha_{t-1}\right)||\boldsymbol{J}_{\boldsymbol{\theta},t-1}(\boldsymbol{x}_{t-1})\Delta\boldsymbol{x}||_2 \\
&\quad + \lambda g\left(\alpha_{t-1},\alpha_t\right)\left(1 - g\left(\alpha_t,\alpha_{t-1}\right)L\right)||\boldsymbol{J}_{\boldsymbol{\theta},t}(\boldsymbol{x}_t)\Delta\boldsymbol{x}||_2, \\
&= -g\left(\alpha_t,\alpha_{t-1}\right)||\hat{\boldsymbol{x}}_{0,t-1}^{\mathcal{W}} - \hat{\boldsymbol{x}}_{0,t-1}||_2 \\
&\quad + g\left(\alpha_{t-1},\alpha_t\right)\left(1 - g\left(\alpha_t,\alpha_{t-1}\right)L\right)||\hat{\boldsymbol{x}}_{0,t}^{\mathcal{W}} - \hat{\boldsymbol{x}}_{0,t}||_2,
\end{aligned}
$$

The first inequality holds because $g\left(\alpha_{t-1},\alpha_t\right) < 0$ and $g\left(\alpha_t,\alpha_{t-1}\right) > 0$. The second inequality holds because $||\boldsymbol{J}_{\boldsymbol{\theta},t-1}(\boldsymbol{x}_{t-1})\boldsymbol{J}_{\boldsymbol{\theta},t}(\boldsymbol{x}_t)\Delta\boldsymbol{x}||_2 \leq ||\boldsymbol{J}_{\boldsymbol{\theta},t-1}(\boldsymbol{x}_{t-1})||_2||\boldsymbol{J}_{\boldsymbol{\theta},t}(\boldsymbol{x}_t)\Delta\boldsymbol{x}||_2 \leq L||\boldsymbol{J}_{\boldsymbol{\theta},t}(\boldsymbol{x}_t)\Delta\boldsymbol{x}||_2$. From Theorem 1, with probability $1 - \dfrac{1}{r_{t-1}}$,

$$||\hat{\boldsymbol{x}}_{0,t-1}^{\mathcal{W}} - \hat{\boldsymbol{x}}_{0,t-1}||_2 \leq \lambda L h(r_{t-1}),$$

with probability $1 - \dfrac{1}{r_t}$,

$$||\hat{\boldsymbol{x}}_{0,t}^{\mathcal{W}} - \hat{\boldsymbol{x}}_{0,t}||_2 \leq \lambda L h(r_t),$$

Thus, from the union of bound, with a probability at least $1 - \dfrac{1}{r_t} - \dfrac{1}{r_{t-1}}$,

$$
\begin{aligned}
||\tilde{x}_t^{\mathcal{W}} - \boldsymbol{x}_t^{\mathcal{W}}||_2 &\leq -\lambda L g\left(\alpha_t,\alpha_{t-1}\right)h(r_{t-1}) + \lambda L g\left(\alpha_{t-1},\alpha_t\right)\left(1 - g\left(\alpha_t,\alpha_{t-1}\right)L\right)h(r_t) \\
&\leq \lambda L\left(-g\left(\alpha_t,\alpha_{t-1}\right) + g\left(\alpha_{t-1},\alpha_t\right)\left(1 - L g\left(\alpha_t,\alpha_{t-1}\right)\right)\right)h(\max\{r_{t-1},r_t\})
\end{aligned}
$$

$\square$

## F  AUXILIARY RESULTS

**Lemma 1.** *Given a unit vector $\boldsymbol{v}_i$ with and $\boldsymbol{\epsilon} \sim \mathcal{N}(\boldsymbol{0}, \boldsymbol{I}_d)$, we have*

$$\mathbb{E}_{\boldsymbol{\epsilon} \sim \mathcal{N}(\boldsymbol{0}, \boldsymbol{I}_d)}[(\boldsymbol{v}_i^T \boldsymbol{\epsilon})^2 / ||\boldsymbol{\epsilon}||_2^2] = \frac{1}{d}.$$

*Proof of Lemma 1.* Because $\boldsymbol{\epsilon} \sim \mathcal{N}(\boldsymbol{0}, \boldsymbol{I}_d)$,

$$\boldsymbol{v}_i^T \boldsymbol{\epsilon} \sim \mathcal{N}(\boldsymbol{v}_i^T \boldsymbol{0}, \boldsymbol{v}_i^T \boldsymbol{I}_d \boldsymbol{v}_i) = \mathcal{N}(\boldsymbol{v}_i^T \boldsymbol{0}, \boldsymbol{v}_i^T \boldsymbol{I}_d \boldsymbol{v}_i) = \mathcal{N}(0, 1), \tag{19}$$

Assume a set of $d$ unit vecotrs $\{v_1, v_2, \ldots, \boldsymbol{v}_i, \ldots, v_d\}$ are orthogonormal and are basis of $\mathbb{R}^d$, similarly, we could show that $\forall j \in [d], X_j := v_j^T \boldsymbol{\epsilon} \sim \mathcal{N}(0, 1)$. Therefore, we could rewrite $(\boldsymbol{v}_i^T \boldsymbol{\epsilon})^2 / ||\boldsymbol{\epsilon}||_2^2$ as:

$$(\boldsymbol{v}_i^T \boldsymbol{\epsilon})^2 / ||\boldsymbol{\epsilon}||_2^2 = \frac{(\boldsymbol{v}_i^T \boldsymbol{\epsilon})^2}{|| \sum_{k=1}^d v_k v_k^T \boldsymbol{\epsilon}||_2^2}, \tag{20}$$

$$= \frac{(\boldsymbol{v}_i^T \boldsymbol{\epsilon})^2}{\sum_{k=1}^d (v_k^T \boldsymbol{\epsilon})^2}, \tag{21}$$

$$= \frac{X_i^2}{\sum_{k=1}^d X_k^2}. \tag{22}$$

Let $Y_i := \frac{X_i^2}{\sum_{j=1}^d X_j^2}$. Because $\forall j \in [d], X_j := v_j^T \boldsymbol{\epsilon} \sim \mathcal{N}(0, 1), \forall j \in [d], Y_j$ has the same distribution. Additionally, $\sum_{j=1}^d Y_j = 1$. So:

$$\mathbb{E}_{\boldsymbol{\epsilon} \sim \mathcal{N}(\boldsymbol{0}, \boldsymbol{I}_d)}[\frac{(\boldsymbol{v}_i^T \boldsymbol{\epsilon})^2}{||\boldsymbol{\epsilon}||_2^2}] = \mathbb{E}[Y_i] = \frac{1}{d}\mathbb{E}[\sum_{j=1}^d Y_j] = \frac{1}{d}.$$

$\square$

**Lemma 2.** *Given a matrix $\boldsymbol{J} \in \mathbb{R}^{d \times d}$ with $\mathrm{rank}(\boldsymbol{J}) = r$. Given $\boldsymbol{x}$ which is uniformly sampled on the unit hypersphere $\mathbb{S}^{d-1}$, we have:*

$$\mathbb{E}_{\boldsymbol{x}}\left[||\boldsymbol{J}\boldsymbol{x}||_2^2\right] = \frac{||\boldsymbol{J}||_F^2}{d}.$$

*Proof of Lemma 2.* Let's define the singular value decomposition of $\boldsymbol{J} = \boldsymbol{U}\boldsymbol{\Sigma}\boldsymbol{V}^T$ with $\boldsymbol{\Sigma} = \mathrm{diag}(\sigma_1, \ldots, \sigma_r, 0 \ldots, 0)$. Therefore, $\mathbb{E}_{\boldsymbol{x}}\left[||\boldsymbol{J}\boldsymbol{x}||_2^2\right] = \mathbb{E}_{\boldsymbol{x}}\left[||\boldsymbol{U}\boldsymbol{\Sigma}\boldsymbol{V}^T\boldsymbol{x}||_2^2\right] = \mathbb{E}_{\boldsymbol{z}}\left[||\boldsymbol{\Sigma}\boldsymbol{z}||_2^2\right]$ where $\boldsymbol{z} := \boldsymbol{V}^T\boldsymbol{x}$ is is uniformly sampled on the unit hypersphere $\mathbb{S}^{d-1}$. Thus, we have:

$$\mathbb{E}_{\boldsymbol{z}}\left[||\boldsymbol{\Sigma}\boldsymbol{z}||_2^2\right] = \mathbb{E}_{\boldsymbol{z}}\left[|| \sum_{i=1}^r \sigma_i \boldsymbol{e}_i^T \boldsymbol{z}||_2^2\right],$$

$$= \mathbb{E}_{\boldsymbol{z}}\left[\sum_{i=1}^r \sigma_i^2 ||\boldsymbol{e}_i^T \boldsymbol{z}||_2^2\right],$$

$$= \sum_{i=1}^r \sigma_i^2 \mathbb{E}_{\boldsymbol{z}}\left[||\boldsymbol{e}_i^T \boldsymbol{z}||_2^2\right] = \frac{||\boldsymbol{J}||_F^2}{d},$$

where $\boldsymbol{e}_i$ is the standard basis with $i$-th element equals to 0. The second equality is because of independence between $\boldsymbol{e}_i^T \boldsymbol{z}$ and $\boldsymbol{e}_j^T \boldsymbol{z}$. The fourth equality is from Lemma 1. $\square$

**Lemma 3.** *Given function $f(\boldsymbol{x}) = ||\boldsymbol{J}\boldsymbol{x}||_2^2$, the lipschitz constant $L_f$ of function $f(\boldsymbol{x})$ is:*

$$L_f = 2||\boldsymbol{J}||_2^2.$$

*Proof of Lemma 3.* The jacobian of $f(\boldsymbol{x})$ is:

$$\nabla_{\boldsymbol{x}} f(\boldsymbol{x}) = 2\boldsymbol{J}^T \boldsymbol{J} \boldsymbol{x},$$

Therefore, the lipschitz constant $L$ follows:

$$L_f = \sup_{\boldsymbol{x} \in \mathbb{S}^{d-1}} ||\nabla_{\boldsymbol{x}} f(\boldsymbol{x})||_2 = 2 \sup_{\boldsymbol{x} \in \mathbb{S}^{d-1}} ||\boldsymbol{J}^T \boldsymbol{J} \boldsymbol{x}||_2 = ||\boldsymbol{J}^T \boldsymbol{J}||_2 = ||\boldsymbol{J}||_2^2$$

$\square$

