# OpenReview forum: "Shallow Diffuse: Robust and Invisible Watermarking through Low-Dimensional Subspaces in Diffusion Models"
_NeurIPS.cc/2024/Workshop/SafeGenAi — SafeGenAi Poster_

### Official Review · Reviewer_5J2v · 2024-10-09
**Comments**

**Rating:** 8
**Confidence:** 4

**Review:**

Summary:

The paper presents a novel watermarking technique that effectively decouples watermark embedding from the diffusion sampling process. By utilizing low-dimensional subspaces, this approach addresses significant challenges faced by existing methods, such as vulnerability to attacks and lack of flexibility, while supporting both server-side and user-side watermarking.

(+) The introduced method is novel and insightful.  This paper offers a fresh perspective on watermarking by leveraging low-dimensional subspaces in diffusion models. This innovative approach not only enhances the robustness and invisibility of the watermarks but also provides a flexible framework that can adapt to different embedding scenarios, addressing key limitations of previous methods.

(+)  The experimental validation is extensive and demonstrates the effectiveness of the Shallow Diffuse method across various datasets and scenarios. The results clearly show superior robustness, consistency, and detectability compared to existing techniques, supported by well-organized tables and figures that enhance the clarity of the findings.

(+) The writing is clear and coherent, effectively conveying the motivation behind the study and the methodology. To further strengthen the paper, it is advisable to enhance the discussion on related works and ensure thorough proofreading to eliminate minor errors. These improvements will significantly enhance the paper's chances for acceptance at the conference.

---

### Official Review · Reviewer_Ugcm · 2024-10-09
**This paper presents Shallow Diffuse, an innovative and flexible watermarking method for robust and invisible protection in both server and user scenarios.**

**Rating:** 7
**Confidence:** 2

**Review:**

The paper introduces Shallow Diffuse, a new watermarking approach aimed at addressing issues related to the misuse of AI-generated content. By utilizing low-dimensional subspaces in diffusion models, the technique separates watermarking from the image generation process, ensuring that the watermark remains both robust and invisible without compromising image quality. Shallow Diffuse stands out by being applicable in both server and user scenarios, offering flexibility not seen in earlier methods. The paper is well-organized, with clear explanations underpinned by strong theoretical analysis and comprehensive experiments, demonstrating the method's superiority in terms of robustness and consistency when compared to existing techniques.

### **Pros**
- **Theoretical guarantees**: One of the standout strengths of Shallow Diffuse is its rigorous theoretical backing. The authors provide provable guarantees regarding the watermark’s consistency and detectability, which is a rarity in the field. They establish theoretical bounds (Theorems 1 and 2) that ensure the watermark’s reliability, which sets this method apart from others that rely more heavily on empirical results. These theoretical foundations help build confidence in the method's general applicability and robustness under various conditions, offering a stronger justification for its use in real-world applications.
- **Novel approach:** Shallow Diffuse advances watermarking by separating it from the image generation process, a departure from traditional methods that incorporate watermarking throughout the sampling process, often degrading image quality. By embedding the watermark within a low-dimensional subspace, it remains hidden in the null space, thus limiting any interference with image clarity. This creative approach significantly reduces distortions and enhances the watermark's invisibility and durability, making it more resistant to typical corruptions and adversarial attacks.
- **Flexibility:** One of Shallow Diffuse's major strengths is its applicability in both server-side and user-side scenarios, making it much more versatile than other methods. While most techniques are limited to a specific scenario, Shallow Diffuse works seamlessly across both, making it practical for a broad range of applications, from protecting media content to tracing the origins of AI-generated images. Its ability to maintain robustness in multiple use cases significantly boosts its practical value.

### **Cons**
- **Dependence on DDIM inversion:** Shallow Diffuse’s reliance on DDIM inversion introduces some error into the watermarking process. Although the authors acknowledge this and provide a well-reasoned explanation, a more detailed investigation into the nature and scale of these errors would improve the paper. Understanding how these errors impact the overall watermarking process, especially in more complex or noisy environments, would be beneficial. Additionally, an exploration of alternative methods to minimize these errors could further strengthen the proposed approach.
- **Missing performance benchmarks:** Although Shallow Diffuse is presented as more efficient than methods like RivaGAN, the paper does not provide detailed data on the computational costs involved in applying this method on a larger scale. Information on how well the technique scales in real-world scenarios, such as when watermarking large volumes of generated images, would be beneficial. An analysis of computational overhead, memory usage, and execution time would help clarify the trade-offs involved in using Shallow Diffuse in practice.
- **Limited exploration of potential downsides:** While the paper focuses on the strengths of Shallow Diffuse, it doesn’t explore its vulnerabilities in sufficient depth. For example, the method’s performance under extreme adversarial conditions or more complex corruption scenarios is not fully explored. Although the robustness is well-demonstrated in standard corruption scenarios, future work could address potential vulnerabilities to more sophisticated adversarial attacks, such as those designed to exploit the specific low-dimensional subspaces used in the watermarking process.

---

### Official Review · Reviewer_kkgt · 2024-10-10
**A new approach of watermark injection and detection for diffusion models**

**Rating:** 6
**Confidence:** 4

**Review:**

Strength:
1) This proposed Shallow Diffuse is new. It provides a watermarking technique that embeds robust and invisible watermarks into diffusion
model outputs.
2) It provides extensive experiments to validate the proposed efforts.

Weakness:
The proposed watermarking method heavily relies on the assumption that the low-dimensional subspace structure in the diffusion model is consistent across different models and datasets. This may limit the generalizability of the technique to models that do not exhibit these properties or to real-world data that doesn't fit well within a low-dimensional subspace.

---

### Official Review · Reviewer_ciDr · 2024-10-12
**This paper introduces Shallow Diffuse, a method for embedding a watermark into the intermediate embedding timestep of the diffusion process. The Jacobian of the PMP is known to be small at these intermediate timesteps, which helps minimize the distortion caused by the watermarks. Shallow Diffuse shows strong robustness and consistency across several benchmarks.**

**Rating:** 6
**Confidence:** 3

**Review:**

**Quality**
The method is sound and supported by a solid theoretical foundation. The inclusion of ablation studies strengthens the claims. However, the motivation for creating a unified method that works in both the user and server scenarios isn’t fully clear to me.

**Clarity**
The paper is generally well-written, with the key background information clearly referenced. The explanations are easy to follow.

**Originality**
This method builds on the idea from Chen et al. (2024), applying it to watermarks using the Tree-Rings Watermarks approach from Wen et al. (2023a). The approach is novel and improves the consistency of watermarked images.

**Significance**
This work contributes to the safer use of GenAI, but the impact is limited by the relatively small performance improvements. Specifically, I don’t fully understand why a unified method is needed for both the user and server scenarios. Wouldn’t it be more effective to use RivaGAN for the user scenario and Gaussian Shading for the server scenario?

**Pros and Cons**
Pros: Novel approach, strong theoretical background, and clear presentation.
Cons: Limited motivation for a unified method and marginal performance gains.

**Additional Questions and Minor Comments**
* In Line 363, it says, "*Gaussian Shading embeds the watermark into $x_T$ , which is not accessible to the user.*" But if we have access to the DDIM Inversion function (which I think we do assume in Shallow Diffuse), can’t we just apply it until we reach ​$x_T$?
* In Line 182, you mention, "*the numerical rank ratio is less than $10^{-2}$,*" but the original reference says "$10^{-1}$." This seems important since it changes whether the watermark's influence is reduced by 1% or 10%.